# Effects of Disinfectants Used for COVID-19 Protection on the Color and Translucency of Acrylic Denture Teeth

Nick Polychronakis [1,*], Aikaterini Mikeli [1], Panos Lagouvardos [2] and Gregory Polyzois [1]

1 Department of Prosthodontics, School of Dentistry, National & Kapodistrian University of Athens, 11527 Athens, Greece
2 Department of Operative Dentistry, School of Dentistry, National & Kapodistrian University of Athens, 11527 Athens, Greece
* Correspondence: nicpolis@dent.uoa.gr; Tel.: +30-210-7461182

**Abstract:** Purpose: This study investigated the color and translucency changes of denture teeth after immersion in disinfectant solutions. Material and Methods: Ten denture teeth (Optostar/Heraeus Kulzer) were immersed in nine different solutions (ethanol 78%, 2-propanol 75%, NaOCl 1%, $H_2O_2$ 0.5%, glutaraldehyde 2.6%, chlorhexidine 0.12%, povidone-iodine 1%, Listerine Naturals, distilled water) for 3 min to 180 min. L*, a* and b* values were measured before and after their immersion with a contact colorimeter (FRU-WR18/Shenzhen Wave Electronics) over a white and black background, and $\Delta E^*_{ab}$, $\Delta E_{00}$, $\Delta TP_{ab}$ and $\Delta TP_{00}$ differences were calculated from baseline measurements. Two-way rmANOVA was used to analyze the data for significant differences among solutions and immersion times at $\alpha = 0.05$. Results: $\Delta E^*_{ab}$ and $\Delta E_{00}$ values were significantly different only across solutions ($p < 0.001$), with mean differences from 0.24 to 1.81 $\Delta E^*_{ab}$ or 0.12 to 0.93 in $\Delta E_{00}$ units. $TP_{ab}$ or $TP_{00}$ translucency parameters showed no significant differences among intervals or solutions ($p > 0.050$). The mean changes ranged from −0.43 to 0.36 $\Delta TP_{ab}$ units, and −0.22 to 0.27 in $\Delta TP_{00}$ units. Conclusions: Most of the solutions had no significant effect on the color of teeth compared to the water group. Chlorhexidine 0.12%, glutaraldehyde 2.6% and Listerine produced significant color changes, especially at 180 min. The translucency of teeth was not affected by the solutions, regardless of the type and immersion time.

**Keywords:** disinfectant solutions; COVID-19; acrylic denture teeth; color; translucency

## 1. Introduction

With more than 630 million confirmed cases and over 6.5 million death cases [1], the severe acute respiratory syndrome coronavirus 2 (SARS-CoV-2) pandemic has not only affected daily life but unavoidably also the dental practice in a long-term manner [2]. In particular, prosthodontists are exposed directly to infection because of the performance of dental procedures on patients, aerosols created during high-speed tooth preparation, exposure to saliva during impressions, contaminated debris from polishing temporary restorations, etc., and indirectly through contact with possibly contaminated saliva from removable prosthetic appliances, impressions and dental stone casts [3–5]. Because of the high virus load in the oral cavity, the detection of the virus in saliva, even in asymptomatic or pre-symptomatic patients, and the fact that human coronaviruses remain infectious on inanimate surfaces at room temperature for extended periods [4,6–8], dental clinicians are obliged to find means to reduce the risk of infection through preventive control procedures.

SARS-CoV-2 is an enveloped, single-stranded RNA virus [8,9]. Due to structural similarities between SARS-CoV and SARS-CoV-2, disinfection media tested against SARS-CoV, such as reagents acting on the outer lipid membrane or on the capsid by denaturing proteins, are expected to have a similar virucidal effect against SARS-CoV-2 [5,6,9]. The inactivation of human coronaviruses on inanimate surfaces over these mechanisms by different types

of widely used biocidal agents has been reported [5,7]. The use of ethanol (78–95%) has reduced the SARS-CoV infectivity by approximately 4 log10 after 30 s of exposure time, similarly to 2-propanol (70–100%), while glutardialdehyde (0.5–2.5%) reduced the viral infectivity of SARS-CoV after a few minutes of exposure time [7]. Moreover, sodium hypochlorite solution 1% demonstrated a similar effect after 1 min, reducing the risk of viral cross-contamination [5].

Since the virus concentration in the saliva reservoir and virus transmission through droplets play an essential role in the COVID-19 pandemic [2,8–10], antiviral mouth rinses could be used in the fight against the transmission of SARS-CoV-2. Hydrogen peroxide has been used in dentistry for over a century [10]. SARS-CoV-2 is vulnerable to oxidation through disruption of the lipid membrane; thus, a rinse containing oxidative agents is expected to reduce the virus load [8–10]. Chlorhexidine is a broad-spectrum antiseptic used in dentistry to treat periodontal disease and is known to have antiviral activity against enveloped viruses [3,9]. While its efficiency in killing SARS-CoV-2 has been controversial, it has been suggested that chlorhexidine rinses could reduce the transmission risk of the virus [3,9–11]. On the other hand, povidone-iodine 1% has antiviral activity against both enveloped and non-enveloped viruses, and in vitro studies have demonstrated virucidal activity against SARS-CoV-2 [8,9,11]. Mouth rinses containing essential oils interfere with the phospholipid bilayer of coronaviruses and could possibly offer antiviral effects against SARS-CoV-2 [9].

Artificial teeth and denture bases are both important components of a removable prosthesis. In terms of esthetics, although the denture base is of great importance to the appearance of the prosthesis, the most important part in the anterior region is the denture teeth. During eating, drinking, mouth-washing or the cleaning procedures of dentures, the artificial teeth are constantly exposed to alkalis, acids and colorants, which interact with their surface, leading to pigmentation [12].

Although many materials can be used in the manufacturing of denture teeth, the preferred ones are thermosetting materials, mainly derived from acrylic acid or acrylate monomers such as methyl-methacrylate [13]. Poly-methyl-methacrylate (PMMA) is the most common material since it is cheap, easily handled, resistant to heat and chemicals, biocompatible, easily colored to match teeth (being transparent) and presents strong adherence to the denture base. Acrylic resin teeth formed of PMMA have low water sorption due to the crosslinking agent included in their composition [14], are resistant to solubility and present a high conversion rate, with low quantities of additional reagents, such as dibenzoyl peroxide, that remain after the reaction and may cause a deterioration in color [15].

Water sorption and solubility are useful indicators of acrylic resins' vulnerability to oral fluids. Low water sorption and low solubility are desired properties for an increase in denture base durability [14]. Acrylic resin teeth demonstrate water sorption abilities, principally because of the inherent nature of resin molecules and their polarity, and, therefore, they tend to absorb water or liquids containing staining agents [16,17]. The water molecules act as plasticizers, causing the expansion of the polymer matrix by separating the polymer chains and jeopardizing the resin's mechanical properties, such as hardness, transverse strength and fatigue limit [18], also exposing the resin to stain penetration and discoloration [19].

As mentioned above, while the use of disinfectants in daily dental practice is unavoidable, their compatibility with dental materials and, moreover, prosthetic materials is essential in order to avoid adverse effects [20]. In particular, the color and translucency of artificial teeth and the denture base material are important properties that should remain unaltered after any disinfection procedures, since any change indicates material aging and damage and, most importantly, a degradation in esthetics that may lead to patient dissatisfaction [21].

A potential adverse event after the immersion of the removable dental prostheses in disinfectants and antiseptics could be a possible alteration of these significant parameters [5,22–25]. Di Fiore et al., in their review, reported color alterations when using NaOCl

depending on the concentration of the medium and the overall exposure time, albeit resulting in acceptable clinical outcomes [5]. On the contrary, the use of ethanol, even in various concentrations (40 and 70%), interferes with the polymer chains, irreversibly deteriorating the surface of the acrylic resin [26]. Color changes were observed also after immersion in povidone-iodine. Silva et al. reported color alterations of denture acrylic resin teeth after repeated immersion cycles in distilled water, 1%, 2% and 5.25% NaOCl, 2% glutaraldehyde and 4% chlorhexidine gluconate, but with no clinical significance [23]. Huang et al. reported significant but clinically acceptable color changes depending on the type of disinfectant used in a 2-year period [25]. Piskin et al. concluded that chemical disinfectants such as NaOCl, sodium perborate, PVP-I, chlorhexidine gluconate and glutaraldehyde affected the color of acrylic denture teeth [24].

While the color, staining susceptibility and translucency of denture teeth immersed in various beverages, food colorants and/or denture cleansers have been investigated [12,17,23,24,27–33], there is a lack of similar studies on the impact of the disinfectants suggested to be used in COVID-19 protocols. Such surface-disinfecting solutions may be chosen by a dental laboratory to disinfect dentures before delivering them to the dentist. They can also be used by the practitioner to disinfect the dentures each time they are removed from the patient's mouth, or by the patient for the daily disinfection of their dentures in a bath. Therefore, we need to determine their effects on the color and translucency of denture teeth for the time used in the protocols. Similarly, mouth disinfectants used by patients on a daily basis may have an effect on denture teeth, which would also be interesting to evaluate.

The aim of this in vitro study was to evaluate the effects of various surface or mouth disinfectants used to prevent the contamination or decrease the load of SARS-CoV-2, on the color and translucency of denture teeth. The null hypothesis of the study was that none of the tested solutions have a significant effect on the color and translucency parameters, with no significant difference from the control medium.

## 2. Materials and Methods

For the study, 90 upper right incisor denture teeth (Optostar, Heraeus Kulzer GmbH, Hanau, Germany) of A3 shade with a rectangular form were assigned randomly to 9 groups of 10 teeth each. Sample size was estimated a priori by G*Power software (G* Power v. 3.1.9.2; University of Kiel) with input parameters $\alpha = 0.05$, $1 - \beta = 0.90$, and an effect size (f) of 0.40. Computations indicated a maximum total size of 90 (10 for each group) with actual power of 0.929 to 0.974.

Each group was immerged in 9 different solutions (Table 1) for 3 min, 60 min and 180 min. Five of the solutions were surface disinfectants and 3 were mouth disinfectants, while distilled water was used as a control. One hundred mL of each solution was poured into 200 mL glass bottles and kept at $24 \pm 1$ °C (surface disinfectants and control) or $37 \pm 1$ °C (mouth disinfectants). All teeth were hydrated in distilled water for 48 h before the baseline measurement, and afterwards the L*, a* and b* color coordinates of teeth in the CIELAB system were measured at the specified time intervals. Measurements were standardized using a silicone mold for the exact positioning of the teeth's labial surfaces on the opening (4 mm) of the measuring colorimeter (FRU-WR18; Shenzhen Wave Optoelectronics Technology Co, Ltd., Shenzhen, China).

**Table 1.** Materials used in the study.

| Material/Lot (Abbrev *) Code | Concentration, Composition, pH | Manufacturer |
|---|---|---|
| Ethanol (ETH)-1 | 78%, pH = 6.8 | |
| 2-Propanol (PROP)-2 | 75%, pH = 7.0 | |
| Sodium hypochlorite (NaOCl)-3 | 1%, pH = 9.5 | |
| Hydrogen peroxide ($H_2O_2$)-4 | 0.5%, pH = 6.0 | |

| Material/Lot (Abbrev *) Code | Concentration, Composition, pH | Manufacturer |
|---|---|---|
| Glutaraldehyde/D20712S (GLUT)-5 | 2.6%, pH = 6.24 | Laboratories ANIOS 59260 Lezennes, France |
| Chlorhexidine gluconate (CHX)-6 | 0,12%, pH = 6.25 | |
| Povidone-iodine (PVP-I)-7 | 1%, pH = 5.8 | |
| Listerine Naturals/230035330 (LIST)-8 | Aqua, sorbitol, propylene glycol, sodium lauryl sulfate, poloxamer 407, sodium saccharine, eucalyptol, benzoic acid, sodium benzoate, methyl salicylate, thymol, menthol, pH = 3.64 | Johnson & Johnson GmbH D-41470 Neuss, DE, Germany |
| Distilled water (WAT)-9 | pH = 7.08 | |
| Optostar (OPT)/1913536681 | Crosslinked PMMA copolymer, inorganic fillers, pigments | Heraeus Kulzer GmbH, Leipziger Str. 263450 Hanau, Germany |

* Abbrev = abbreviation.

Three measurements were taken over a white and black background. The white and black background was achieved by painting white and black the inner surfaces of two identical silicone molds with white (ANNIE no02, Paris, France) and black (ANNIE no75, Paris, France) opaque nail varnish. For the measurements, the teeth were removed from the solution, excess water was carefully absorbed on a soft paper tissue, and they were measured and then immersed again in the solution. Overall color changes of teeth from baseline values were calculated in $\Delta E^*_{ab}$ and $\Delta E_{00}$ units, at all time intervals, using the following formulas [34,35].

$$\Delta E^*ab = ([L^*1 - L^*2]2 + [a^*1 - a^*2]2 + [b^*1 - b^*2]2)1/2, \text{ and}$$

$$\Delta E_{00} = ([\Delta L'/KLSL]2 + [\Delta C'/KCSC]2 + [\Delta H'/KHSH]2 + RT[\Delta C'/KCSC][\Delta H'/KHSH])1/2$$

Factors $\Delta L'$, $\Delta C'$, $\Delta H'$ represent differences in lightness, chroma and hue. KL, KC and KH are weighted factors. SL, SC and SH are averaging factors for lightness, chroma and hue. RT is an overall correction factor based on differences in hue and chroma. The weighted parametric factors KL, KC and KH were set to 2, 1 and 1.

The translucency of teeth at all time intervals was based on the translucency parameter (TPab) formula [36],

$$TP_{ab} = ([L^*b - L^*w]2 + [a^*b - a^*w]2 + [b^*b - b^*w]2)1/2$$

where b and w denote black and white backgrounds. The range of TP can be 0 to 100, meaning that if a material is completely opaque, its TP is zero [36].

Recently, another formula has been suggested based on the CIEDE2000 color difference model [37],

$$TP_{00} = ([\Delta L'/KLSL]2 + [\Delta C'/KCSC]2 + [\Delta H'/KHSH]2 + RT[\Delta C'/KCSC][\Delta H'/KHSH])1/2$$

where $\Delta L'$, $\Delta C'$ and $\Delta H'$ are differences in teeth in terms of lightness, chroma and hue, over a black and white background. This formula, as reported by the authors, provides better data for estimating and visualizing translucency than the TPab formula, and for this reason, we also calculated the $TP_{00}$ values.

Differences in translucency ($\Delta TP$) before and after immersion in disinfectant solutions for each time interval were also calculated as TP changes from baseline measurements with both formulas ($\Delta TP_{ab}$ and $\Delta TP_{00}$).

Two-way repeated-measures ANOVA was used to analyze the data for significant differences among solutions or among immersion times, either for color coordinates or TP,

at a level of significance $\alpha = 0.05$. The IBM-SPSS statistical package v. 25 was used for the analysis (IBM Corp, Armonk, NY, USA).

## 3. Results

### 3.1. L*, a* and b* Parameters

L*, a* and b* values of teeth before and after their immersion in the solutions for 3 min, 60 min and 180 min are shown in Table 2. The table shows small numeric differences at baseline among teeth, with a mean of 3.11, 0.36 and 1.31 units for L*, a* and b* coordinates, respectively, which remained almost the same till the end of the experiment.

**Table 2.** Values of L*, a* and b* color parameters of teeth at all time intervals of immersion in the investigated solutions.

| Solution | L* 0 | 3′ | 60′ | 180′ | a* 0 | 3′ | 60′ | 180′ | b* 0 | 3′ | 60′ | 180′ |
|---|---|---|---|---|---|---|---|---|---|---|---|---|
| 1-ETH | 59.29 ± 0.31 | 59.62 ± 0.34 | 59.53 ± 0.38 | 59.76 ± 0.3 | 1.47 ± 0.1 | 1.44 ± 0.08 | 1.38 ± 0.12 | 1.25 ± 0.1 | 8.58 ± 0.47 | 8.65 ± 0.34 | 8.68 ± 0.38 | 8.60 ± 0.35 |
| 2-PROP | 58.87 ± 0.81 | 58.76 ± 0.65 | 58.89 ± 0.66 | 58.9 ± 0.72 | 1.41 ± 0.14 | 1.34 ± 0.1 | 1.26 ± 0.17 | 1.32 ± 0.14 | 8.48 ± 0.45 | 8.46 ± 0.44 | 8.47 ± 0.44 | 8.51 ± 0.45 |
| 3-NaOCl | 58.59 ± 0.88 | 58.46 ± 0.64 | 58.22 ± 1.38 | 58.67 ± 0.86 | 1.55 ± 0.28 | 1.45 ± 0.35 | 1.48 ± 0.33 | 1.47 ± 0.32 | 8.83 ± 0.42 | 8.79 ± 0.46 | 8.75 ± 0.61 | 8.74 ± 0.44 |
| 4-H$_2$O$_2$ | 58.83 ± 1.01 | 58.43 ± 0.47 | 58.84 ± 0.88 | 58.55 ± 0.75 | 1.34 ± 0.13 | 1.38 ± 0.18 | 1.3 ± 0.15 | 1.27 ± 0.14 | 8.40 ± 0.36 | 8.49 ± 0.37 | 8.46 ± 0.37 | 8.46 ± 0.35 |
| 5-GLUT | 58.4 ± 0.89 | 58.02 ± 0.53 | 57.15 ± 2.07 | 57.82 ± 1.13 | 1.70 ± 0.12 | 1.59 ± 0.11 | 1.69 ± 0.19 | 1.62 ± 0.12 | 7.59 ± 0.25 | 7.56 ± 0.25 | 7.95 ± 0.5 | 7.62 ± 0.23 |
| 6-CHX | 56.15 ± 1.19 | 57.09 ± 0.62 | 57.24 ± 1.01 | 56.38 ± 1.35 | 1.66 ± 0.21 | 1.64 ± 0.2 | 1.56 ± 0.11 | 1.55 ± 0.14 | 7.52 ± 0.57 | 7.46 ± 0.47 | 7.52 ± 0.4 | 7.74 ± 0.46 |
| 7-PVP-I | 57.76 ± 1.00 | 57.42 ± 0.77 | 58.00 ± 0.93 | 58.00 ± 0.88 | 1.48 ± 0.15 | 1.57 ± 0.09 | 1.58 ± 0.1 | 1.61 ± 0.09 | 7.93 ± 0.25 | 8.12 ± 0.54 | 8.05 ± 0.4 | 8.10 ± 0.41 |
| 8-LIST | 57.43 ± 0.97 | 58.34 ± 0.58 | 57.41 ± 1.41 | 57.94 ± 1.19 | 1.67 ± 0.16 | 1.65 ± 0.1 | 1.61 ± 0.11 | 1.69 ± 0.09 | 7.88 ± 0.21 | 8.00 ± 0.18 | 8.23 ± 0.51 | 8.08 ± 0.32 |
| 9-WAT | 59.12 ± 0.41 | 58.91 ± 0.35 | 59.19 ± 0.45 | 59.17 ± 0.43 | 1.45 ± 0.11 | 1.44 ± 0.11 | 1.43 ± 0.08 | 1.40 ± 0.10 | 8.63 ± 0.39 | 8.61 ± 0.40 | 8.58 ± 0.41 | 8.59 ± 0.40 |

### 3.2. $\Delta E^*_{ab}$ and $\Delta E_{00}$ Diffcerences

Calculated $\Delta E^*ab$ and $\Delta E_{00}$ values are shown in Table 3. Statistical analysis showed significant differences across solutions ($p < 0.001$), but no differences across time intervals ($p_{ab} = 0.395$, $p_{00} = 0.216$) or with time–solution interaction ($p_{ab} = 0.181$, $p_{00} = 0.185$). The differences across solutions are shown also in Table 3, based on post-hoc pairwise comparisons with Bonferroni correction. $\Delta E^*_{ab}$ values across solutions varied from 0.25 to 1.25 units at 3 min, reaching 1.69 and 1.81 units at 60 and 180 min. $\Delta E_{00}$ values ranged from 0.12 to 0.64 units at 3 min, increasing to 0.93 at 180 min. The differences among solutions in $\Delta E^*_{ab}$ values were from 1.01 to 1.65 units and in $\Delta E_{00}$ values from 0.52 to 0.82 units.

**Table 3.** $\Delta E^*_{ab}$ and $\Delta E_{00}$ values of teeth from baseline measurements, at three different time intervals of immersion in the investigated solutions.

| Solution | $\Delta E^*_{ab}$ 3 Min | 60 Min | 180 Min | +Stats | $\Delta E_{00}$ 3 Min | 60 Min | 180 Min | +Stats |
|---|---|---|---|---|---|---|---|---|
| 1-ETH | 0.57 ± 0.39 | 0.57 ± 0.34 | 0.71 ± 0.45 | abc | 0.35 ± 0.28 | 0.39 ± 0.25 | 0.51 ± 0.28 | abc |
| 2-PROP | 0.64 ± 0.49 | 0.61 ± 0.50 | 0.75 ± 0.49 | abc | 0.38 ± 0.21 | 0.42 ± 0.22 | 0.41 ± 0.24 | abc |
| 3-NaOCl | 0.85 ± 0.73 | 1.00 ± 0.91 | 0.90 ± 0.80 | abc | 0.47 ± 0.32 | 0.52 ± 0.41 | 0.45 ± 0.36 | abc |
| 4-H$_2$O$_2$ | 0.79 ± 0.47 | 0.26 ± 0.21 | 0.49 ± 0.44 | ab | 0.40 ± 0.20 | 0.18 ± 0.12 | 0.28 ± 0.18 | ab |
| 5-GLUT | 0.73 ± 0.61 | 1.81 ± 2.06 | 1.15 ± 1.24 | bc | 0.41 ± 0.31 | 0.93 ± 1.03 | 0.58 ± 0.58 | bc |
| 6-CHX | 1.17 ± 0.76 | 1.65 ± 0.89 | 1.69 ± 1.40 | c | 0.64 ± 0.32 | 0.83 ± 0.41 | 0.92 ± 0.69 | c |
| 7-PVP-I | 0.96 ± 0.87 | 0.71 ± 0.82 | 1.13 ± 0.94 | abc | 0.52 ± 0.41 | 0.42 ± 0.37 | 0.62 ± 0.42 | bc |
| 8-LIST | 1.25 ± 0.94 | 1.36 ± 1.33 | 1.32 ± 1.08 | bc | 0.61 ± 0.42 | 0.72 ± 0.67 | 0.66 ± 0.51 | bc |
| 9-WAT | 0.25 ± 0.13 | 0.16 ± 0.06 | 0.22 ± 0.12 | a | 0.12 ± 0.06 | 0.11 ± 0.04 | 0.15 ± 0.06 | a |

+Stats columns show the significant ($p < 0.05$) marginal differences among solutions. No significant difference across time intervals or time–solution interaction was found. Same letters in the cells indicate no difference at $\alpha = 0.05$ significance level. Differences were based on post-hoc pairwise comparisons using Bonferroni correction.

### 3.3. Translucency Parameters and Differences

Translucency parameters ($TP_{ab}$ and $TP_{00}$) of teeth in the solutions, at all time intervals, are shown in Table 4. Group means were found at baseline to be approximately 2.00 units in $TP_{ab}$ and approximately 1.2 units in $TP_{00}$ values. The differences from baseline ($\Delta TP_{ab}$ and $\Delta TP_{00}$) are presented in Table 5. Repeated-measures ANOVA on $\Delta TP$ values showed no significant differences among time intervals ($p_{ab} = 0.669$, $p_{00} = 0.806$), among solutions ($p_{ab} = 0.701$, $p_{00} = 0.127$) or with time–solution interaction ($p_{ab} = 0.767$, $p_{00} = 0.263$). The mean changes at 3 min ranged from $-0.43$ to $0.36$ $\Delta TP_{ab}$ units and $-0.25$ to $0.27$ $\Delta TP_{00}$ units.

**Table 4.** Translucency parameters of teeth ($TP_{ab}$ and $TP_{00}$) before (0 min) and after immersion in the solutions for 3 min, 60 min and 180 min.

| Code-Abbrev | $TP_{ab}$ 0 Min | 3 Min | 60 Min | 180 Min | $TP_{00}$ 0 Min | 3 Min | 60 Min | 180 Min |
|---|---|---|---|---|---|---|---|---|
| 1-ETH | 1.09 ± 0.34 | 0.67 ± 0.57 | 0.98 ± 0.13 | 0.99 ± 0.26 | 0.95 ± 0.15 | 0.75 ± 0.25 | 1.07 ± 0.17 | 0.99 ± 0.22 |
| 2-PROP | 0.97 ± 0.20 | 1.09 ± 0.40 | 0.72 ± 0.96 | 0.91 ± 0.75 | 1.03 ± 0.21 | 1.00 ± 0.23 | 0.82 ± 0.37 | 0.95 ± 0.29 |
| 3-NaOCl | 0.77 ± 0.18 | 0.81 ± 0.42 | 1.27 ± 0.21 | 0.89 ± 0.42 | 0.87 ± 0.23 | 0.65 ± 0.14 | 0.91 ± 0.13 | 0.78 ± 0.25 |
| 4-H$_2$O$_2$ | 0.66 ± 1.11 | 0.99 ± 0.88 | 0.86 ± 1.06 | 0.89 ± 0.89 | 0.85 ± 0.53 | 0.88 ± 0.52 | 0.83 ± 0.69 | 0.78 ± 0.51 |
| 5-GLUT | 2.80 ± 0.82 | 3.16 ± 0.82 | 2.80 ± 0.55 | 2.46 ± 0.94 | 1.91 ± 0.39 | 2.11 ± 0.40 | 2.03 ± 0.28 | 1.85 ± 0.44 |
| 6-CHX | 2.84 ± 0.68 | 3.01 ± 0.56 | 2.76 ± 2.05 | 2.93 ± 1.44 | 2.09 ± 0.44 | 2.19 ± 0.30 | 2.18 ± 0.28 | 2.12 ± 0.64 |
| 7-PVP-I | 2.28 ± 0.85 | 2.47 ± 0.63 | 2.98 ± 0.77 | 2.61 ± 0.99 | 1.61 ± 0.43 | 1.88 ± 0.35 | 1.74 ± 0.37 | 1.95 ± 0.51 |
| 8-LIST | 2.35 ± 0.87 | 2.27 ± 0.96 | 2.24 ± 1.08 | 2.18 ± 1.05 | 1.83 ± 0.48 | 1.59 ± 0.57 | 1.57 ± 0.58 | 1.49 ± 0.56 |
| 9-WAT | 0.81 ± 0.48 | 0.80 ± 0.13 | 0.80 ± 0.14 | 0.71 ± 0.38 | 0.97 ± 0.17 | 0.96 ± 0.12 | 0.94 ± 0.09 | 0.91 ± 0.14 |

**Table 5.** $\Delta TP_{ab}$ and $\Delta TP_{00}$ values of teeth from baseline measurements at three different time intervals in the investigated solutions.

| Code-Abbrev | $\Delta TP_{ab}$ 3 Min | 60 Min | 180 Min | $\Delta TP_{00}$ 3 Min | 60 Min | 180 Min |
|---|---|---|---|---|---|---|
| 1-ETH | −0.43 ± 0.72 | −0.11 ± 0.36 | −0.11 ± 0.45 | −0.20 ± 0.24 | 0.12 ± 0.16 | 0.04 ± 0.21 |
| 2-PROP | 0.12 ± 0.43 | −0.25 ± 0.83 | −0.06 ± 0.7 | −0.03 ± 0.35 | −0.21 ± 0.46 | −0.08 ± 0.32 |
| 3-NaOCl | 0.03 ± 0.38 | 0.5 ± 0.24 | 0.12 ± 0.42 | −0.22 ± 0.15 | 0.03 ± 0.19 | −0.09 ± 0.33 |
| 4-H$_2$O$_2$ | 0.34 ± 0.65 | 0.2 ± 1.33 | 0.23 ± 1.13 | 0.03 ± 0.46 | −0.01 ± 0.82 | −0.07 ± 0.59 |
| 5-GLUT | 0.36 ± 1.05 | 0.00 ± 1.00 | −0.35 ± 1.28 | 0.20 ± 0.36 | 0.12 ± 0.33 | −0.06 ± 0.55 |
| 6-CHX | 0.17 ± 0.91 | −0.08 ± 2.47 | 0.09 ± 1.81 | 0.10 ± 0.51 | 0.09 ± 0.50 | 0.03 ± 0.87 |
| 7-PVP-I | 0.2 ± 0.97 | 0.7 ± 0.98 | 0.34 ± 1.06 | 0.27 ± 0.46 | 0.14 ± 0.44 | 0.35 ± 0.58 |
| 8-LIST | −0.08 ± 0.25 | −0.11 ± 0.72 | −0.17 ± 0.71 | −0.25 ± 0.19 | −0.26 ± 0.35 | −0.35 ± 0.33 |
| 9-WAT | −0.01 ± 0.15 | −0.03 ± 0.21 | −0.06 ± 0.24 | −0.01 ± 0.15 | −0.03 ± 0.21 | −0.06 ± 0.24 |

Note: No significant differences were found among solutions, time intervals or time–solution interaction with both TP formulas ($p > 0.05$).

### 3.4. Correlation Analyses

Correlation analysis between $\Delta TP$ and $\Delta E$ values showed small correlation coefficients (r) for most of the solution pairs, $-0.48$ to $0.20$, for both formulas.

## 4. Discussion

The null hypothesis that none of the tested solutions would have a significant effect on the color of the teeth was rejected, since many of them affected the color coordinates. Moreover, the hypothesis that no significant differences would exist between the solutions and the control medium (water) was also rejected, since differences in the $\Delta E^*_{ab}$ and $\Delta E_{00}$ values among many pairs were significant. On the contrary, regarding the translucency, the null hypothesis was accepted since none of the solutions affected the $TP_{ab}$ or $TP_{00}$ values and no differences were detected between the various disinfectant solutions and the control group.

In this study, water demonstrated the lowest effect on L*, a* and b* values (0.25 $\Delta E^*_{ab}$ units) from the beginning to the end of the experiment, meaning that water had the

lowest effect on tooth color among all solutions. However, Silva et al. reported that the immersion of teeth in water caused greater color changes than 2% NaOCl, povidone-iodine, glutaraldehyde and chlorhexidine gluconate disinfectant solutions [23]. This disagreement may be attributed to many differences between the two studies. Silva et al. evaluated discs composed of PMMA in interpenetrating polymer network (IPN) structures and without immersing them in water before baseline measurements [23]. This means that the changes from the absorption of water, which usually take place in the first few days of immersion (56 h–190 h of 80% uptake), dependent always on the material [38], were probably included in the measurements. Moreover, because the more homogeneous a material is, the less water it absorbs, due to a lower degree of porosity [39,40], either the IPN structure or disc manufacturing resulted in a less homogeneous surface that absorbed more water and caused more color changes than the experimental solutions.

This study indicated also that glutaraldehyde 5 solution affected the color of teeth significantly more than most of the solutions, including the control. These results are consistent with those of Piskin et al., who reported statistically significant differences in $\Delta E^*_{ab}$ values between the water (control) and the glutaraldehyde group [24]. In this study, the greatest change was found when teeth were immersed in glutaraldehyde for 60 min, but in the study of Piskin et al., the greatest change was noted with sodium hypochlorite 5.25% [24]. However, glutaraldehyde's effect was also high. Glutaraldehyde owes its biocidal action to its ability to react with functional groups of proteins such as amine, thiol and phenol groups (protein crosslinking reagents). In aqueous solutions, it may have many different forms (monomeric or polymeric) depending on the solution temperature, pH and concentration, and because these forms are in equilibrium, there is no agreement as to which form is the most reactive one in the crosslinking process [41]. For this reason, we cannot give a definite explanation for the changes in color found in teeth immersed in glutaraldehyde solution. To address this question, a specific and well-designed study is needed. However, we may assume that the color changes may result from the reaction of monomeric forms of glutaraldehyde with some of the free radicals on tooth surfaces or/and the precipitation on tooth surfaces of its polymeric forms, since both can alter the optical characteristics of teeth.

Chlorhexidine is a broad-spectrum antibacterial agent acting against Gram-positive and -negative organisms and fungus, by disrupting their outer membranes. Chlorhexidine gluconate 0.12% was found to have the second-greatest discoloration effect among all disinfectants, significantly higher than the control solution and $H_2O_2$, increasing with time. Similar results concerning water and chlorhexidine solutions were reported by Patel et al., although direct comparison of the results is not possible, due to differences in solution concentrations, immersion times and tooth types [42]. Silva et al. found a statistically significant difference between water and chlorhexidine solution after 90 immersion cycles [23]. The increase in $\Delta E^*ab$ with time was attributed to the staining ability of chlorhexidine as a result of the local precipitation reaction between the cationic chlorhexidine molecules attached to the teeth or dental materials and the chromogens contained in the saliva, after the intake of food or beverages [43]. In fact, three different mechanisms have been suggested: (1) the Maillard reaction, which is the reaction between reducing sugars and proteins catalyzed by chlorhexidine, resulting in the formation of colored pigments (melanoidins); (2) protein denaturation by chlorhexidine and the formation of organic yellow–brown ferric sulfides; and (3) chlorhexidine and pigment interaction from colored food and beverages [44,45]. In the present study, none of these mechanisms offer a reasonable explanation since no chromogens were introduced to the solutions. However, since cationic chlorhexidine molecules are gradually attached to the tooth surfaces, this may build a layer that impairs the normal optical properties of teeth. A similar mechanism is used by the recently introduced chlorhexidine products to reduce its discoloration effect on teeth. They incorporate a polyvinylpyrrolidone/vinyl acetate copolymer (PVP-VA), which forms a transparent film on teeth or material surfaces, protecting them from discoloration [21].

Listerine is a mouth disinfectant that owes its bactericidal, antivirus and antifungal action to volatile oils such as eucalyptol, thymol and menthol. The solution used in this study also contained ethanol as a binder for oils, sodium lauryl sulfate as a surfactant and benzoic acid with its salt sodium benzoate as preservatives. This solution was another disinfectant that affected the tooth color significantly more than water ($p > 0.05$). Ertürk-Avunduk et al. found also that a Listerine solution with a low pH affected significantly the color of a bulk-fill resin composite. Since acids may catalyze the ester groups in methyl-methacrylate groups, it is possible that the acidity of Listerine (pH = 3.64) may be responsible for the minor degradation in the polymeric matrix of teeth and the subsequent discoloration [46].

The povidone-iodine solution, due to its dark red color, was expected to change significantly the color of the teeth immersed in it. Surprisingly, although the teeth presented high variation in color, no significant difference from the control group was observed. It must be noted that all measurements were performed on the teeth immediately after the immersion period and without any further procedure performed on their surfaces, except careful drying. Povidone-iodine is a skin disinfectant owing its antibacterial (gram positive or negative), antivirus and antifungal action to the slow release of iodine from povidone or polyvinylpyrrolidone (PVP), which is a water-soluble polymer. Since it is a strong disinfectant against many bacteria and viruses, it is safe in terms of color stability to be used for the short-term disinfection of denture teeth.

In this study, although statistically significant differences were found among the solutions in their effects on the color of teeth, the values of the changes were not higher than 1.81 or 0.93 in $\Delta E^*_{ab}$ or $\Delta E_{00}$ units (Table 3). The values of 1.2 and 2.7 have been recommended as the 50%:50% perceptibility and acceptability thresholds in $\Delta E^*_{ab}$ units [47], and, respectively, the values of 1.3 and 2.25 in $\Delta E_{00}$ units [48]. Therefore, although the values may be above the perceptibility level, they remain within a clinically acceptable range.

The translucency of artificial teeth should be close to that of natural teeth, since it is of great importance for denture esthetics [33]. Changes in translucency indicate possible modifications of surface roughness and/or the material's mass, leading to the higher diffusion and lower reflectance of light on their surfaces [49]. Salas et al. reported that CIEDE2000 50:50% TPT was 0.62 and TAT was 2.62, with corresponding CIELAB values of 1.33 and 4.43, respectively. In the present study, the mean change of teeth in translucency from baseline values was in the range of 0.09 to 0.71 $\Delta TP_{ab}$ units or 0.05 to 0.31 $\Delta TP_{00}$ units, changes that are therefore not perceptible [37], with no significant differences among solutions or immersion times and no correlation of $\Delta E^*$ values with $\Delta TP$ values. Divnic-Resnik et al., in their study, found significant differences in the $\Delta TP_{ab}$ of materials immersed in mouth disinfectants, which were solution- but not material-dependent, with a significant solution–material interaction. The longer period of immersion in the solutions (28 days) and the use of composite resins as the investigated material are probably the reasons for their results and the differences between the two studies [45].

The present study answers the question of whether disinfectants have or do not have a significant effect on the color and translucency of denture teeth, if they are used for short periods of time, as in recent COVID-19 disinfection protocols. Ethanol 78%, 2-propanol 75%, sodium hypochlorite 1%, hydrogen peroxide 0.5% and povidone-iodine 1% do not affect the color and translucency of denture teeth. However, glutaraldehyde 2.6%, chlorhexidine gluconate 0.12% and Listerine Naturals do affect their color. The later are all included in the category of mouth disinfectants and may be used for long periods of time by the individual. In this study, the longest period was 180 min, which corresponds to 90 days of 1 min rinse twice a day, simulating only the time period, since no rinses with water or complete dentures were used. This could be considered one of the limitations of this study, although the study was not intended to investigate the long-term effects of mouth disinfectants. Another limitation of this study is that only one type of artificial teeth was investigated. It is possible that teeth with different compositions and structure could be more vulnerable than the type that we have used. Another study with more tooth types but fewer disinfectants,

perhaps those found to affect the color of teeth more potently, would be useful. In this study, measurements of color were based on the results of a contact colorimeter with an opening of 4 mm. This opening is considered small and therefore all measurements suffer equally from the edge-loss effect. However, besides the fact that a larger diameter could have resulted in measurements with greater error due to surface curvature, the selected teeth were central incisors with a rather flat surface and were embedded in white silicone, minimizing the edge-loss effect, as Yu et al. reported in their study, embedding the teeth in plasticine [50].

## 5. Conclusions

Based on the findings of this in vitro study, the following conclusions can be drawn:

1. Most of the solutions had no significant effect on the color of teeth compared to the control (water) group. Water affected only slightly the color coordinates of teeth, resulting in a change of 0.16 to 0.25 $\Delta E^*_{ab}$ units or 0.11 to 0.15 $\Delta E^*_{00}$ units.
2. Three of the solutions, glutaraldehyde 2.6%, chlorhexidine 0.12% and Listerine Naturals, affected significantly the color of the teeth compared to the water group.
3. The highest color change was found when teeth were immersed in chlorhexidine 0.12% or glutaraldehyde 2.6% at 60 min and 180 min, reaching the maximum of 1.81 $\Delta E^*_{ab}$ units or 0.93 $\Delta E_{00}$ units.
4. Immersion of teeth in the solutions had no significant effect on their translucency, regardless of the type or immersion time. The changes in translucency were always below the 50%:50% perceptibility threshold.

**Author Contributions:** Conceptualization, N.P.; methodology, N.P. and P.L.; software, P.L. and N.P.; validation, G.P., N.P. and P.L.; formal analysis, P.L.; investigation, N.P. and A.M.; resources, N.P. and A.M.; data curation, P.L. and N.P.; writing—original draft preparation, N.P., P.L., A.M. and G.P.; writing—review and editing, N.P., P.L., A.M. and G.P.; visualization, N.P. and P.L.; supervision, G.P. and P.L.; project administration, N.P. All authors have read and agreed to the published version of the manuscript.

**Funding:** This research received no external funding.

**Institutional Review Board Statement:** Not applicable.

**Informed Consent Statement:** Not applicable.

**Conflicts of Interest:** The authors declare no conflict of interest.

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
