# Peer review of "Effects of Disinfectants Used for COVID-19 Protection on the Color and Translucency of Acrylic Denture Teeth"

_prosthesis, doi:10.3390/prosthesis5010009_

Round 1
Reviewer 1 Report
This study investigates color and translucency changes of denture teeth after immersion in disinfectant solutions. The study is well conducted and the protocol well explained. The choice of tested disinfectants is pertinent. The authors may find some remarks below concerning the way to present the results.
However, I don’t understand the point of view chosen by the authors with regards to Covid virus. What’s the clinical sense of the immersion in those disinfectants ? Does it mean that the practitioner will rinse all dentures in disinfectants to avoid infected projections ? Would it be some advice to give to the patients who feel like disinfecting their dentures ? Or some laboratory recommendation when delivering new dentures ?
About the result section and statistics.
Table 2 and L*, a* and b* parameters. I understand that there is no difference after immersion, whatever the time and the disinfectant. That’s reassuring. In those conditions, is it worthy to present all the table ? Maybe add a conclusion line at the end of the paragraph for clinical meaning.
ΔΕ*ab and ΔΕ00 differences. The table is unclear. The way to show the solutions presenting statically significant differences is unusual. Why not use small superscript letters ?
Again, explain if the variations in ab and E00 is clinically meaningful.
The authors discussed the mechanisms of color changing with as much literature as they found. Yet, these are mainly hypothesis.
Reviewer 2 Report
Line 34: The authors wrote:
“pandemic has not only affected the daily life but unavoidably also the dental practice in a long-term aspect.”
Please support this sentence with a reference. A suggestion: https://doi.org/10.23736/S2724-6329.21.04632-5
The FRU-WR18 comes in three variations: 40mm, 8mm, 4mm. Which one did you use?
Please specify. Edit: this was specified at the end of the discussion, but it should also be specified in materials and methods.
Lines 106-8:
The authors wrote:
“On the contrary the use of ethanol even in various concentrations (40 and 70%) interfere with the polymer chains irreversibly deteriorating the surface of the acrylic resin.”
Please support this sentence with a reference. The reviewer suggest a recent review that should be cited in the current paper for several reasons: 1) the effect of ethanol on resin materials and subsequent staining 2) the effect of colorants on resin materials 3) the relationship between different resin compositions and different liquid compositions.
The reference is: Paolone G, Formiga S, De Palma F, Abbruzzese L, Chirico L, Scolavino S, Goracci C, Cantatore G, Vichi A. Color stability of resin-based composites: Staining procedures with liquids-A narrative review. J Esthet Restor Dent. 2022 Sep;34(6):865-887. doi: 10.1111/jerd.12912. Epub 2022 Apr 9. PMID: 35396818.
Lines 161: constants KLSL, H, C shall be declared (what they are and the values). Were they 1?
Line 166: The authors wrote:
“A greater TP means a greater translucency but also less opacity ”. Authors could consider removing “ but also less opacity” while it is redundant.
Line 171: The authors wrote: “This formula according to the authors”
It seems that this is a personal opinion of the authors of the current reviewed article. Please rephrase.
Lines 182-4: please remove these lines.
The authors wrote: “the longest period was 180 min which corresponds to 90 days of 1 min rinse twice a day”
In the limitations, the authors should mention that in 90 days, there is no rinsing, no use of the dentures, and factors that may affect color stability.
The authors should also specify why two formulas (ΔE*ab and ΔE*00) were used. The latter is more indicated for dental color difference detection.
The last item in the conclusions is related to the “translucency perceptibility threshold”. This threshold has not been mentioned in the text. In the paper, the authors cited the values od color change perceptibility and acceptability thresholds (which were not cited in the conclusions) but no values for translucency thresholds were cited. Both thresholds, results, and clinical implications shall be mentioned in the discussion and conclusions.
Round 2
Reviewer 1 Report
Answers to the comments were not satisfactory. I don't see the scientific interest or the issue raised in the introduction.
Scientific forms, such as the way to present statistical differences, follows general rules known and accepted by the scientific community. We have to stick to them, it is a way to assure dissemination of knowledge.
Round 3
Reviewer 1 Report
Thank you for your modifications.
I think your article will be of clinical interest to many practitioners.